# FedComLoc: Communication-Efficient Distributed Training of Sparse and Quantized Models

## Abstract

Federated Learning (FL) has garnered increasing attention due to its unique characteristic of allowing heterogeneous clients to process their private data locally and interact with a central server, while being respectful of privacy. A critical bottleneck in FL is the communication cost. A pivotal strategy to mitigate this burden is *Local Training*, which involves running multiple local stochastic gradient descent iterations between communication phases. Our work is inspired by the innovative Scaffnew algorithm of Mishchenko et al. (2022), which has considerably advanced the reduction of communication complexity in FL. We introduce FedComLoc (Federated Compressed and Local Training), integrating practical and effective compression into Scaffnew to further enhance communication efficiency. Extensive experiments, using the popular TopK compressor and quantization, demonstrate its prowess in substantially reducing communication overheads in heterogeneous settings.

## 1 Introduction

Privacy concerns and limited computing resources on edge devices often make centralized training impractical, where all data is gathered in a data center for processing. In response to these challenges, Federated Learning (FL) has emerged as an increasingly popular framework (McMahan et al., 2016; Kairouz et al., 2019). In FL, multiple clients perform computations locally on their private data and exchange information with a central server. This process is typically framed as an empirical risk minimization problem (Shalev-Shwartz & Ben-David, 2014):

$$\min_{x \in \mathbb{R}^d} \left[ f(x) := \frac{1}{n} \sum_{i=1}^{n} f_i(x) \right], \tag{ERM}$$

where $f_i$ represents the local objective for client $i$, $n$ is the total number of clients, and $x$ is the model to be optimized. Our primary objective is to solve the problem (ERM) and deploy the optimized global model to all clients. For instance, $x$ might be a neural network trained in an FL setting. However, a considerable bottleneck in FL are communication cost, particularly with large models.

To mitigate these costs, FL often employs *Local Training* (LT), a strategy where local parameters are updated multiple times before aggregation (Povey et al., 2014; Moritz et al., 2016; McMahan et al., 2017; Li et al., 2020; Haddadpour & Mahdavi, 2019; Khaled et al., 2019; 2020; Karimireddy et al., 2020; Gorbunov et al., 2020; Mitra et al., 2021). However, there is a lack of theoretical understanding regarding the effectiveness of LT methods. The recent introduction of Scaffnew by Mishchenko et al. (2022) marked a substantial advancement, as this algorithm converges to the exact solution with accelerated complexity, in convex settings.

Another approach to reducing communication costs is through compression (Haddadpour et al., 2021; Condat et al., 2022; Yi et al., 2024). In centralized training, one often aims to learn a sparsified model for faster training and communication efficiency (Dettmers & Zettlemoyer, 2019; Kuznedelev et al., 2023). Dynamic pruning strategies like gradual magnitude pruning (Gale et al., 2019) and RigL (Evci et al., 2020) are common. But in FL, the effectiveness of these model sparsification methods based on thresholds remains unclear. The work by Babakniya et al. (2023) considers FL sparsity mask concepts, showing promising results.

Quantization is another efficient model compression technique (Han et al., 2021; Bhalgat et al., 2020; Shin et al., 2023), though its application in heterogeneous settings is limited. Gupta et al. (2022) introduced FedAvg with Kurtosis regularization (Chmiel et al., 2020) in FL.

Furthermore, studies such as Haddadpour et al. (2021); Condat et al. (2022) have theoretical convergence guarantees for unbiased estimators with restrictive assumptions. As this work employs the biased TopK compressor these are unsuitable in this case.

Thus, we tackle the following question:

*Is it possible to design an efficient algorithm combining accelerated local training with compression techniques, such as quantization and Top-K, and validate its efficiency empirically on popular FL datasets?*

Our method for investigating this question consists of two steps. Firstly, we design an algorithm, termed FedComLoc, which integrates general compression into ScaffNew, an accelerated local training algorithm. Secondly, we empirically validate FedComLoc for popular compression techniques (TopK and quantization) on popular FL datasets (FedMNIST and FedCIFAR10).

We were able to answer this question affirmatively with the following contributions:

- We have developed a communication-efficient method FedComLoc for distributed training, specifically designed for heterogeneous environments. This method integrates general compression techniques and is motivated by previous theoretical insights.

- We proposed three variants of our algorithm addressing several key bottlenecks in FL: FedComLoc-Com addresses communication costs from client to server, FedComLoc-Global addresses communication costs from server to client and FedComLoc-Local addresses limited computational resources on edge devices.

- We conducted detailed comparisons and ablation studies, validating the effectiveness of our approach. These reveal a considerable reduction in communication and, in certain cases, an enhancement in training speed in number of communication rounds. Furthermore, we demonstrated that our method outperforms well-established baselines in terms of training speed and communication costs.

## 2 RELATED WORK

### 2.1 LOCAL TRAINING

The evolution of LT in FL has been profound and continuous, transitioning through five distinct generations, each marked by considerable advancements from empirical discoveries to reductions in communication complexity. The pioneering FedAvg algorithm (McMahan et al., 2017) represents the first generation of LT techniques, primarily focusing on empirical evidence and practical applications (Povey et al., 2014; Moritz et al., 2016; McMahan et al., 2017). The second generation of LT methods consists in solving (ERM) based on homogeneity assumptions such as bounded gradients[1] (Li et al., 2020) or limited gradient diversity[2] (Haddadpour & Mahdavi, 2019). However, the practicality of such assumptions in real-world FL scenarios is debatable and often not viable (Kairouz et al., 2019; Wang et al., 2021).

Third-generation methods made fewer assumptions, demonstrating sublinear (Khaled et al., 2019; 2020) or linear convergence up to a neighborhood (Malinovsky et al., 2020) with convex and smooth functions. More recently, fourth-generation algorithms like Scaffold (Karimireddy et al., 2020), S-Local-GD (Gorbunov et al., 2020), and FedLin (Mitra et al., 2021) have gained popularity. These algorithms effectively counteract client drift and achieve linear convergence to the exact solution under standard assumptions. Despite these advances, their communication complexity mirrors that of GD, i.e. $\mathcal{O}(\kappa \log \epsilon^{-1})$, where $\kappa := L/\mu$ denotes the condition number.

The most recent Scaffnew algorithm, proposed by Mishchenko et al. (2022), revolutionizes the field with its accelerated communication complexity $\mathcal{O}(\sqrt{\kappa} \log \epsilon^{-1})$. This seminal development estab-

---

[1]There exists $c \in \mathbb{R}$ s.t. $\|\nabla f_i(x)\| \leq c$ for $1 \leq i \leq d$.
[2]There exists $c \in \mathbb{R}$ s.t. $\frac{1}{n} \sum_{i=1}^{n} \|\nabla f_i(x)\|^2 \leq c\|\nabla f(x)\|^2$.

---

**Algorithm 1** FedComLoc

---

1: stepsize $\gamma > 0$, probability $p > 0$, initial iterate $x_{1,0} = \cdots = x_{n,0} \in \mathbb{R}^d$, initial control variates $h_{1,0}, \ldots, h_{n,0} \in \mathbb{R}^d$ on each client such that $\sum_{i=1}^{n} h_{i,0} = 0$, number of iterations $T \geq 1$, compressor $\mathrm{C}(\cdot) \in \{\mathrm{Top}K(\cdot), \mathrm{Q_r}(\cdot), \cdots\}$
2: **server:** flip a coin, $\theta_t \in \{0, 1\}$, $T$ times, where $\mathrm{Prob}(\theta_t = 1) = p$     $\diamond$ Decide when to skip communication
3: send the sequence $\theta_0, \ldots, \theta_{T-1}$ to all workers
4: **for** $t = 0, 1, \ldots, T-1$ **do**
5:     sample clients $\mathcal{S} \subseteq \{1, 2, 3, \ldots, n\}$
6:     **in parallel on all workers** $i \in \mathcal{S}$ **do**
7:         FedComLoc-Local: local compression $- g_{i,t}(x_{i,t}) = g_{i,t}(\mathrm{C}(x_{i,t}))$
8:         $\hat{x}_{i,t+1} = x_{i,t} - \gamma(g_{i,t}(x_{i,t}) - h_{i,t})$     $\diamond$ Local gradient-type step adjusted via the local control variate $h_{i,t}$
9:         FedComLoc-Com: uplink compression $- \hat{x}_{i,t+1} = \mathrm{C}(\hat{x}_{i,t+1})$
10:        **if** $\theta_t = 1$ **then**
11:            $x_{i,t+1} = \frac{1}{n} \sum_{i=1}^{n} \hat{x}_{i,t+1}$     $\diamond$ Average the iterates (with small probability $p$)
12:            FedComLoc-Global: downlink compression $- x_{i,t+1} = \mathrm{C}(x_{i,t+1})$
13:        **else**
14:            $x_{i,t+1} = \hat{x}_{i,t+1}$     $\diamond$ Skip communication
15:        **end if**
16:        $h_{i,t+1} = h_{i,t} + \frac{p}{\gamma}(x_{i,t+1} - \hat{x}_{i,t+1})$     $\diamond$ Update the local control variate $h_{i,t}$
17:    **end local updates**
18: **end for**

---

lishes LT as a communication acceleration mechanism for the first time, positioning Scaffnew at the forefront of the fifth generation of LT methods with accelerated convergence. Further enhancements to Scaffnew have been introduced, incorporating aspects like variance-reduced stochastic gradients (Malinovsky et al., 2022), personalization (Yi et al., 2023), partial client participation (Condat et al., 2023), asynchronous communication (Maranjyan et al., 2022), and an expansion into a broader primal–dual framework (Condat & Richtárik, 2023). This latest generation also includes the 5GCS algorithm (Grudzień et al., 2023), with a different strategy where the local steps are part of an inner loop to approximate a proximity operator. Our proposed FedComLoc algorithm extends Scaffnew by incorporating pragmatic compression techniques, such as sparsity and quantization, resulting in even faster training measured by the number of bits communicated.

## 2.2 MODEL COMPRESSION IN FEDERATED LEARNING

Model compression in the context of FL is a burgeoning field with diverse research avenues, particularly focusing on the balance between model efficiency and performance. Jiang et al. (2022) innovated in global pruning by engaging a single, powerful client to initiate the pruning process. This strategy transitions into a collaborative local pruning phase, where all clients contribute to an adaptive pruning mechanism. This involves not just parameter elimination, but also their reintroduction, integrated with the standard FedAvg framework (McMahan et al., 2016). However, this approach demands substantial local memory for tracking the relevance of each parameter, a constraint not always feasible in FL settings.

Addressing some of these challenges, Huang et al. (2022) introduced an adaptive batch normalization coupled with progressive pruning modules, enhancing sparse local computations. These advancements, however, often do not fully address the constraints related to computational resources and communication bandwidth on the client side. Our research primarily focuses on magnitude-based sparsity pruning. Techniques like gradual magnitude pruning (Gale et al., 2019) and RigL (Evci et al., 2020) have been instrumental in dynamic pruning strategies. However, their application in FL contexts remains relatively unexplored. The pioneering work of Babakniya et al. (2023) extends the concept of sparsity masks in FL, demonstrating noteworthy outcomes.

Quantization is another vital avenue in model compression. Seminal works in this area include Han et al. (2021), Bhalgat et al. (2020), and Shin et al. (2023). A major advance has been made by Gupta et al. (2022), who combined FedAvg with Kurtosis regularization (Chmiel et al., 2020). We are looking to go even further by integrating accelerated LT with quantization techniques.

However, a gap exists in the theoretical underpinnings of these compression methods. Research by Haddadpour et al. (2021) and Condat et al. (2022) offers theoretical convergence guarantees for unbiased estimators, but these frameworks are not readily applicable to common compressors like Top-K sparsifiers. In particular, CompressedScaffnew (Condat et al., 2022) integrates an unbiased compression mechanism in Scaffnew, that is based on random permutations. But due to requiring shared randomness it is not practical. Linear convergence has been proved when all functions $f_i$ are strongly convex.

To the best of our knowledge, no other compression mechanism has been studied in Scaffnew, either theoretically or empirically, and even the mere convergence of Scaffnew in nonconvex settings has not been investigated either. Our goal is to go beyond the convex setting and simplistic logistic regression experiments and to study compression in Scaffnew in realistic nonconvex settings with large datasets such as Federated CIFAR and MNIST. Our integration of compression in Scaffnew is heuristic but backed by the findings and theoretical guarantees of CompressedScaffnew in the convex setting, which shows a twofold acceleration with respect to the conditioning $\kappa$ and the dimension $d$, thanks to LT and compression, respectively.

## 3 PROPOSED ALGORITHM FedComLoc

### 3.1 SPARSITY AND QUANTIZATION

Let us define the sparsifying $\mathrm{Top}K(\cdot)$ and quantization $\mathrm{Q_r}(\cdot)$ operators.

**Definition 3.1.** Let $x \in \mathbb{R}^d$ and $K \in \{1, 2, \ldots, d\}$. We define the sparsifying compressor $\mathrm{Top}K : \mathbb{R}^d \to \mathbb{R}^d$ as:

$$\mathrm{Top}K(x) := \arg\min_{y \in \mathbb{R}^d} \{\|y - x\| \mid \|y\|_0 \leq K\},$$

where $\|y\|_0 := |\{i : y_i \neq 0\}|$ denotes the number of nonzero elements in the vector $y = (y_1, \cdots, y_d)^\mathsf{T} \in \mathbb{R}^d$. In case of multiple minimizers, $\mathrm{Top}K$ is chosen arbitrarily.

**Definition 3.2.** For any vector $x \in \mathbb{R}^d$, with $x \neq \mathbf{0}$ and a number of bits $r > 0$, its binary quantization $\mathrm{Q_r}(x)$ is defined componentwise as

$$\mathrm{Q_r}(x) = (\|x\|_2 \cdot \mathrm{sgn}(x_i) \cdot \xi_i(x, 2^r))_{1 \leq i \leq d},$$

where $\xi_i(x, 2^r)$ are independent random variables. Let $y_i := \frac{|x_i|}{\|x\|_2}$. Then their probability distribution is given by

$$\xi_i(x, 2^r) = \begin{cases} \lceil 2^r y_i \rceil / 2^r & \text{with proba. } 2^r y_i - \lfloor 2^r y_i \rfloor; \\ \lfloor 2^r y_i \rfloor / 2^r & \text{otherwise.} \end{cases}$$

If $x = \mathbf{0}$, we define $\mathrm{Q_r}(x) = \mathbf{0}$.

The distributions of the $\xi_i(x, r)$ minimize variance over distributions with support $\{0, 1/r, \ldots, 1\}$, ensuring unbiasedness, i.e. $\mathbb{E}[\xi_i(x, r)] = |x_i| / \|x\|_2$. This definition is based on an equivalent one in Alistarh et al. (2017).

### 3.2 INTRODUCTION OF THE ALGORITHMS

FedComLoc (Algorithm 1) is an adaptation of Scaffnew, with modifications for compression. $\mathrm{Top}K(\cdot)$ is used as the default compression technique for simplicity, although quantization is equally applicable. We examine three distinct variants in our ablations:

- FedComLoc-Com: This variant addresses the communication bottleneck. It focuses on compressing the uplink network weights transmitted from each client to the server. This setup is adopted as our default setting.

| Top-K | 100% | 10% | 30% | 50% | 70% | 90% |
|---|---|---|---|---|---|---|
| Accuracy | 0.9758 | 0.9374 | 0.9654 | 0.9699 | 0.9745 | 0.9748 |
| Decrease | - | 3.94% | 1.07% | 0.61% | 0.13% | 0.10% |

Table 1: Test accuracy for various Top-K ratios.

| | $\alpha = 0.1$ | $\alpha = 0.3$ | $\alpha = 0.5$ | $\alpha = 0.7$ | $\alpha = 0.9$ | $\alpha = 1.0$ |
|---|---|---|---|---|---|---|
| K =100% | 0.9623 | 0.9686 | 0.9731 | 0.9758 | 0.9768 | 0.9735 |
| K =10% | 0.8681 | 0.9124 | 0.9331 | 0.9374 | 0.9441 | 0.9382 |
| K =50% | 0.9597 | 0.9635 | 0.9671 | 0.9699 | 0.9706 | 0.9719 |

Table 2: Test accuracy score for various Dirichlet factors $\alpha$ and sparsity ratios.

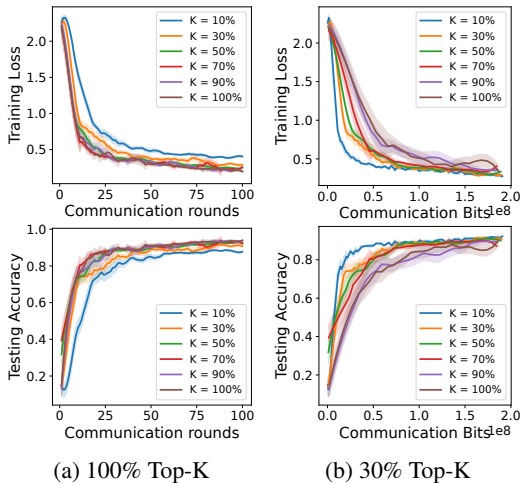

(a) 100% Top-K     (b) 30% Top-K

Figure 1: Performance outcomes for various Top-K ratios.

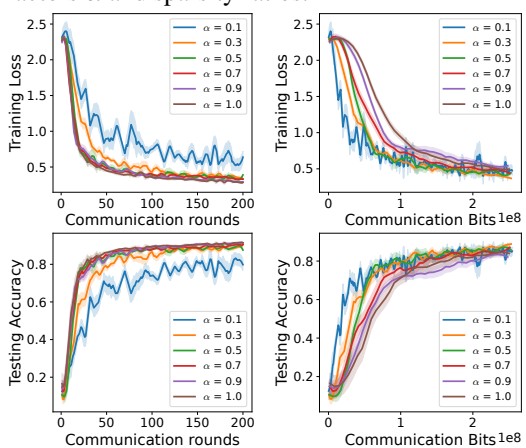

(a) Density ratio $K = 100\%$.    (b) Density ratio $K = 10\%$.

Figure 2: Training loss and test accuracy for a density ratio of $K = 10\%$.

- **FedComLoc-Local**: Here, the local model is compressed during each training step. This addresses limited computational power and resources available to each client.

- **FedComLoc-Global**: Here, the server model is compressed before sending it to the clients. This variant is tailored for FL situations where downloading the model from the server is costly e.g. due to network bandwidth constraints.

## 4 EXPERIMENTS

**Baselines.** Our evaluation comprises three distinct aspects. Firstly, we conduct experiments to assess the impact of compression on communication costs. FedComLoc is assessed for varying sparsity and quantization ratios. Secondly, we compare FedComLoc-Com with FedComLoc-Local and FedComLoc-Global. Thirdly, we explore the efficacy of FedComLoc against non-accelerated local training methods, including FedAvg (McMahan et al., 2016), its Top-K sparsified counterpart sparseFedAvg, and Scaffold (Karimireddy et al., 2020).

**Datasets.** Our experiments are conducted on FedMNIST (LeCun, 1998) and FedCI-FAR10 (Krizhevsky et al., 2009) with the data processing framework FedLab (Zeng et al., 2023). For FedMNIST, we employ MLPs with three fully-connected layers, each coupled with a ReLU activation function. For FedCIFAR10, we utilize CNNs with two convolutional layers and three fully-connected layers. Comprehensive statistics for each dataset, details on network architecture and training specifics can be found in Appendix A.

**Heterogeneous Setting.** We explore different heterogeneous settings. Similar to (Zhang et al., 2023; Yi et al., 2024), we create heterogeneity in data by using a Dirichlet distribution, which assigns each client a vector indicating class preferences. This vector guides the unique selection of labels and images for each client until all data is distributed. The Dirichlet parameter $\alpha$ indicates the level of non-identical distribution. We also include a visualization of this distribution for the CIFAR10 dataset in Appendix B.1.1.

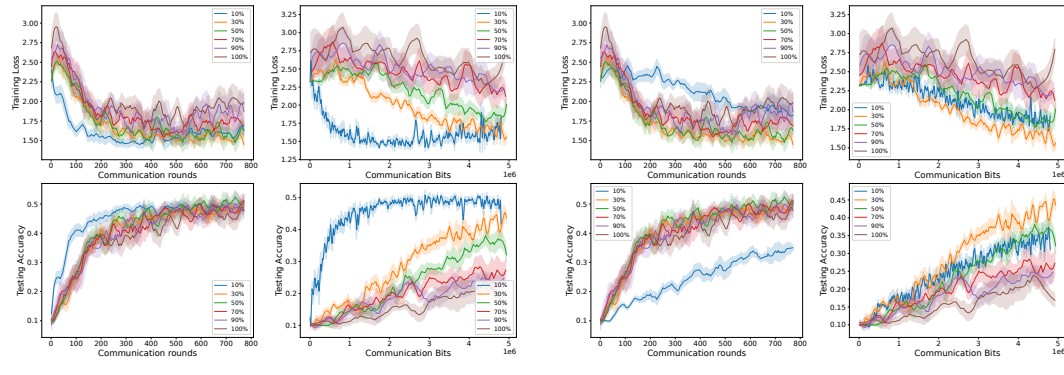

Tuned Stepsize.                                   Fixed Stepsize of 0.01.

Figure 3: CNN Performance on the FedCIFAR10 Dataset. For the left most columns, the step size was optimized for each density ratio $K$. For the two rightmost columns, a fixed stepsize of 0.01 is used. This is the maximum feasible step size which ensures convergence across all configurations.

**Default Configuration.** In the absence of specific clarifications, we adopt the Dirichlet factor $\alpha = 0.7$. To balance both communication and local computation costs, we use $p = 0.1$, resulting in an average of 10 local iterations per communication round. The learning rate is chosen by conducting a grid search over the set $\{0.005, 0.01, 0.05, 0.1, 0.5\}$. With communication costs being of most interest, our study employs FedComLoc-Com as the default strategy. The experiments are run for 2500 communication rounds for the CNN on FedCIFAR10 and 500 rounds for the MLP on FedMNIST. Furthermore, the dataset is distributed across 100 clients from which 10 are uniformly chosen to participate in each global round.

Furthermore, in our Definition 3.1 of TopK, $K$ is the number of nonzero parameters. However, we will rather specify the enforced density ratio, i.e. the ratio of nonzero parameters. For instance, specifying $K = 30\%$ means retaining 30% of parameters.

### 4.1 TOP-K SPARSITY RATIOS

This section investigates the effects of different sparsity rations by investigating TopK ratios on FedMNIST. The outcomes can be found in Table 1. Notably, $K = 10\%$ in TopK yields an accuracy of 0.9374, merely 3.94% lower than the 0.9758 unsparsified baseline. Remarkably, a 70% sparsity level ($K = 30\%$) attains commendable performance, with only a 1.07% accuracy reduction, alongside a 70% reduction in communication costs. Furthermore, from the communication bits depicted in Figure 1 it is evident that sparsity yields faster convergence, the more so with increased sparsity (smaller $K$).

### 4.2 DATA HETEROGENEITY/DIRICHLET FACTORS

This subsection aims to assess the impact of varying degrees of data heterogeneity on FedMNIST. Hence, an analysis of the Dirichlet distribution factor $\alpha$ is presented, exploring the range of values $\alpha \in \{0.1, 0.3, 0.5, 0.7, 0.9, 1.0\}$. Remember that a lower $\alpha$ means increased heterogeneity. Alongside, we examine the influence of different TopK factors, specifically $10\%, 50\%$ and $100\%$. The results are shown in Table 2. Figure 2 reports training loss and test accuracy for a sparsity ratio of 90% ($K = 10\%$). Additionally, round-wise visualizations for $K = 50\%$ and $K = 100\%$ (non-sparse) are presented in Figure 11 in the Appendix.

Key observations from our study include:

a) When examining the effects of the heterogeneity degree $\alpha$ (as seen in each column of Table 2), we observe that sparsity performance is influenced by heterogeneity degrees. For instance, $\alpha = 0.1$ results in a relative performance drop of 9.79% from an unsparsified to a sparsified model with $K = 10\%$. In contrast, for $\alpha = 0.3$, this drop is 5.80%, and for $\alpha = 1.0$, it is 3.63%. Interestingly, for commonly used heterogeneity ratios in literature ($\alpha = 0.3, 0.5, 0.7$), the performance drop does

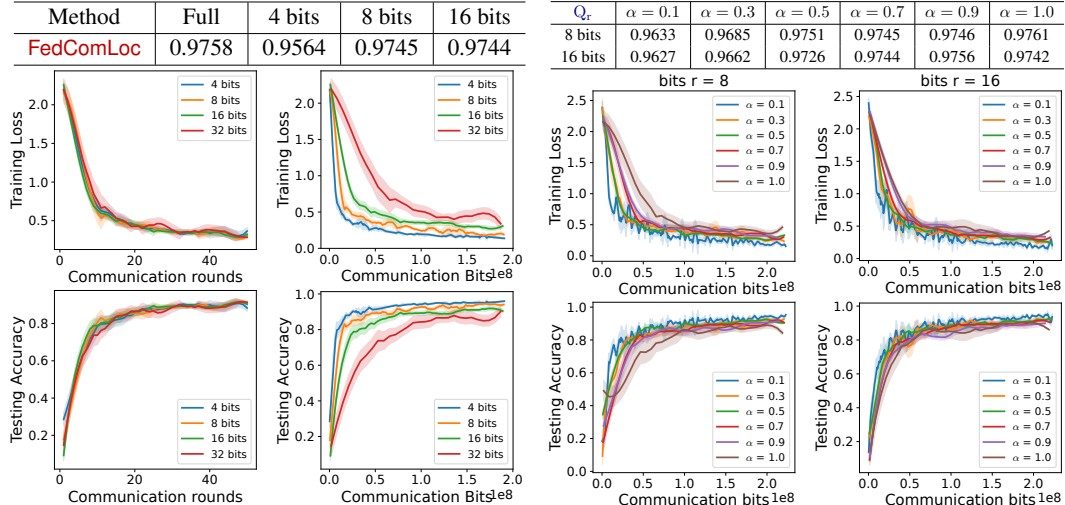

| Method | Full | 4 bits | 8 bits | 16 bits |
|---|---|---|---|---|
| FedComLoc | 0.9758 | 0.9564 | 0.9745 | 0.9744 |

| $Q_r$ | $\alpha = 0.1$ | $\alpha = 0.3$ | $\alpha = 0.5$ | $\alpha = 0.7$ | $\alpha = 0.9$ | $\alpha = 1.0$ |
|---|---|---|---|---|---|---|
| 8 bits | 0.9633 | 0.9685 | 0.9751 | 0.9745 | 0.9746 | 0.9761 |
| 16 bits | 0.9627 | 0.9662 | 0.9726 | 0.9744 | 0.9756 | 0.9742 |

Figure 5: FedComLoc employing $Q_r(\cdot)$. The number of quantization bits $r$ is set to $r \in \{4, 8, 16, 32\}$.

Figure 6: Data heterogeneity ablations for Fed-ComLoc utilizing $Q_r(\cdot)$ with number of bits $r$ either 8 or 16. The same results plotted over the number of communication rounds can be found in Figure 13 in the Appendix.

not decrease substantially when moving from $\alpha = 0.3$ to $\alpha = 0.5$, or from $\alpha = 0.5$ to $\alpha = 0.7$, unlike the shift from $\alpha = 0.1$ to $\alpha = 0.3$.

b) Focusing on the rows of Table 2, we find that lower sparsity ratios are more sensitive to heterogeneous distributions. In particular, observe that with $K = 10\%$, the absolute performance improvement from $\alpha = 0.1$ to $\alpha = 1$ is 7.01%. However, for $K = 50\%$, this improvement is only 1.22%.

c) It should be noted that each method was run with a fixed learning rate without scheduling, and the maximum communication round was set to 1000. Previous studies suggest that higher sparsity ratios require more communication rounds in centralized settings (Kuznedelev et al., 2023). This phenomenon was also observed in our FL experiments. Therefore, there is the potential for performance enhancement through sufficient model rounds and adaptive learning rate adjustments, especially for methods with higher sparsity.

### 4.3 CNNs on FedCIFAR10

This section repeats the experiments for CIFAR10 and a Convolutional Neural Network (CNN). We explored a range of stepsizes ($\gamma \in \{0.005, 0.01, 0.05, 0.1|\}$). Further information is provided in Appendix A. The CIFAR10 results, which involve optimizing a Convolutional Neural Network (CNN), are presented in Figure 3 for both tuned and a fixed step size. Observe the accelerated convergence of sparsified models in terms of communicated bits when the step size is tuned. Interestingly, a sparsity of 90% ($K = 10\%$) shows faster convergence in terms of communication rounds (as shown in the first column), suggesting the potential for enhanced training efficiency in sparsified models. For a fixed step size (the two rightmost columns of Figure 3) and $K = 10\%$, one can observe slower convergence compared to other configurations. This indicates that sparsity training requires more data and benefits from either increased communication rounds or a larger initial stepsize. This aligns with recent similar findings in the centralized setting (Kuznedelev et al., 2023).

### 4.4 QUANTIZATION

This section explores using quantization $Q_r$ for compression with the number of bits, $r$, set to $r \in \{4, 8, 16, 32\}$ on FedMNIST. This approach aligns with the methodologies outlined in Alistarh et al. (2017). The results after 1000 communication rounds are illustrated in Figure 5. Our

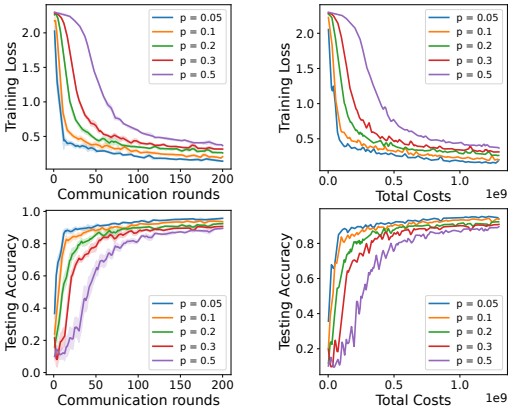

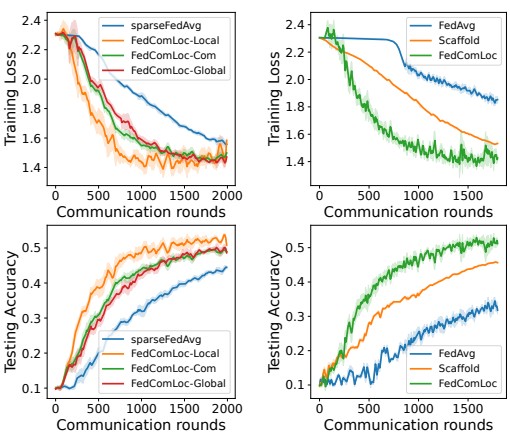

Figure 7: Loss and test accuracy over communication rounds and total costs. Total costs are a combined measurement of both communication costs and local computation cost. A communication round has unit cost while a local training round has cost $\tau$. In a realistic FL system, $\tau$ is typically much less than 1, as the primary bottleneck is often communication and hence we set $\tau = 0.01$.

Figure 8: Comparison among FedAvg, Scaffold, FedDyn, FedComLoc-Local, FedComLoc-Com, and FedComLoc-Global. First column: $K = 50\%$; second column: $K = 100\%$ (no sparsity).

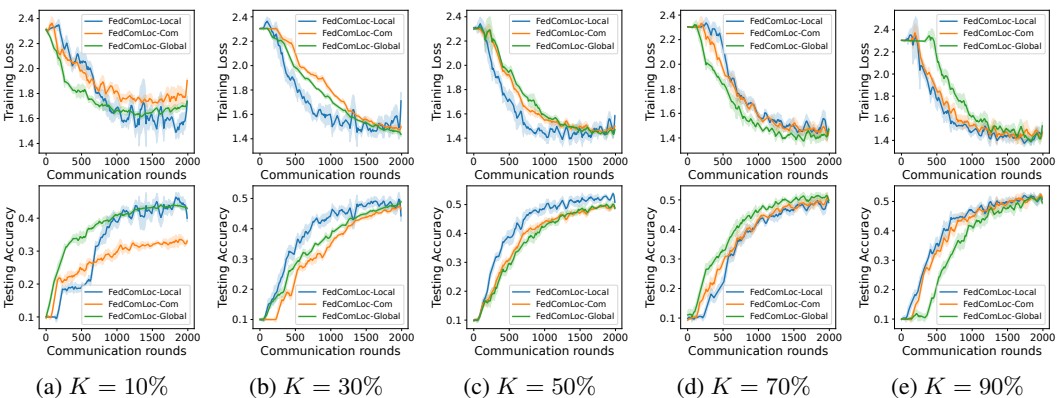

(a) $K = 10\%$     (b) $K = 30\%$     (c) $K = 50\%$     (d) $K = 70\%$     (e) $K = 90\%$

Figure 9: Sparsity ablation studies of FedComLoc-Local, FedComLoc-Com, and FedComLoc-Global on FedCIFAR10 and tuned stepsizes.

analysis reveals that quantization offers superior performance compared to TopK-style sparsity. For instance, with 16-bit quantization corresponding to a 50% reduction in communication cost, the performance decrease is a mere $0.14\%$, Furthermore, Figure 6 shows outcomes for different degrees of data heterogeneity. These findings demonstrate that quantization reduces communication at minor performance tradeoff while also exhibiting only minor sensitivity to data heterogeneity. The Appendix gives further results for both FedMNIST (section B.2.1) and FedCIFAR10 (section B.2.2).

## 4.5 NUMBER OF LOCAL ITERATIONS

This section explores the performance impact of varying the expected number of local iterations on FedMNIST. The expected number of local iterations is $1/p$ where $p$ is the communication probability. Hence, we investigate the influence of $p$ ranging from $p \in \{0.05, 0.1, 0.2, 0.3, 0.5\}$. Furthermore, $K = 30\%$ is used. The results are presented in Figure 7. A key finding is that more local training rounds (i.e. smaller $p$) not only accelerate convergence but can also improve the final performance.

### 4.6 FEDCOMLOC VARIANTS

In this section, we compare FedComLoc-Local, FedComLoc-Com, and FedComLoc-Global on Fed-CIFAR10. The findings are illustrated in Figure 9. Observe that at high levels of sparsity (indicated by a small $K$ in Top$K$), FedComLoc-Com underperforms the other algorithms. This could be attributed to the heterogeneous setting of our experiment: each client's model output is inherently skewed towards its local dataset. When this is coupled with extreme Top$K$ sparsification, more bias is introduced, which adversely affects performance. Conversely, at low sparsity (e.g. $K = 90\%$), FedComLoc-Com surpasses FedComLoc-Global. In addition, we observe that sparsity during local training (i.e. FedComLoc-Local) tends to yield better results. One possible explanation is that due to the local data bias the communication bandwidth between client and server might be crucial. Remember that FedComLoc-Local had no communication compression while both FedComLoc-Com and FedComLoc-Global do. Further FedMNIST results are shown in the Appendix.

### 4.7 FEDAVG AND SCAFFOLD

In this section, the performance of FedComLoc is compared with baselines in form of FedAvg (McMahan et al., 2016) and Scaffold (Karimireddy et al., 2020) on FedCIFAR10. Furthermore, a sparsified version of FedAvg is employed, termed as sparseFedAvg. For sparseFedAvg a learning rate of $0.1$ is used, whereas for FedComLoc, a lower rate of $0.05$ is utilized. The outcomes of this analysis are depicted in Figure 8. The left part illustrates the performance of compressed models. We observe notably faster convergence for FedComLoc-type methods in comparison with sparseFedAvg despite the lower learning rate. The right part of the figure compares FedAvg with Scaffold, devoid of sparsity, using an identical learning rate of $0.005$. This uniform rate ensures that each method achieves satisfactory convergence over a sufficient number of epochs. Here again, faster convergence is demonstrated with FedComLoc.

## 5 CONCLUSION AND FUTURE WORK

This paper advances the field of FL by tackling one of its main challenges, namely its high communication cost. Building on the accelerated Scaffnew algorithm (Mishchenko et al., 2022), we introduced FedComLoc. This novel approach blends the practical compression techniques of model sparsity and quantization into the efficient local training framework. Our extensive experimental validation shows that FedComLoc preserves computational integrity while notably cutting communication costs.

Future research could explore the reduction of internal variance in stochastic gradient estimation, akin to the approach described in Malinovsky et al. (2022). The FedComLoc-Global algorithm we propose offers potential for obtaining a sparsified model suitable for deployment. Additionally, investigating the development of an efficiently sparsified deployed model extensively presents an intriguing avenue for further study.

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

CONTENTS

## A    EXPERIMENTAL DETAILS

### A.1    DATASETS AND MODELS

Our research primarily focuses on evaluating the effectiveness of our proposed methods and various baselines on widely recognized FL datasets. These include Federated MNIST (FedMNIST) and Federated CIFAR10 (FedCIFAR10), which are benchmarks in the field. The use of the terms FedMNIST and FedCIFAR10 is intentional to distinguish our federated training approach from the centralized training methods typically used with MNIST and CIFAR10. The MNIST dataset consists of 60,000 samples distributed across 100 clients using a Dirichlet distribution. For this dataset, we employ a three-layer Multi-Layer Perceptron (MLP) as our default model. CIFAR10, also comprising 60,000 samples, is utilized in our experiments to conduct various ablation studies. The default setting for our FedCIFAR10 experiments is set with 10 clients. The model chosen for CIFAR10 is a Convolutional Neural Network (CNN) consisting of 2 convolutional layers and 3 fully connected layers (FCs). The network architecture is chosen in alignment with (Zeng et al., 2023).

### A.2    TRAINING DETAILS

Our experimental setup involved the use of NVIDIA A100 or V100 GPUs, allocated based on their availability within our computing cluster. We developed our framework using PyTorch version 1.4.0 and torchvision version 0.5.0, operating within a Python 3.8 environment. The FedLab framework (Zeng et al., 2023) was employed for the implementation of our code. For the FedMNIST dataset, we established the default number of global iterations at 500, whereas for the FedCIFAR10 dataset, this number was set at 2500. We conducted a comprehensive grid search for the optimal learning rate, exploring values within the range of $[0.005, 0.01, 0.05, 0.1]$. Our intention is to make the code publicly available upon the acceptance of our work.

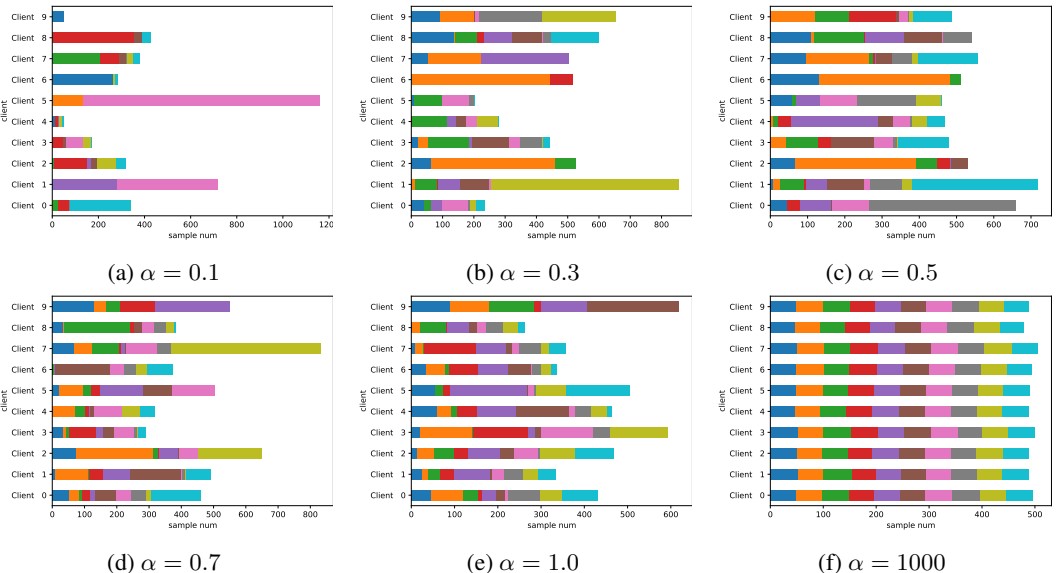

(a) $\alpha = 0.1$      (b) $\alpha = 0.3$      (c) $\alpha = 0.5$

(d) $\alpha = 0.7$      (e) $\alpha = 1.0$      (f) $\alpha = 1000$

Figure 10: Data distribution with different Dirichlet factors on CIFAR10 distributed over 100 clients

## B    COMPLEMENTARY EXPERIMENTS

### B.1    EXPLORING HETEROGENEITY

#### B.1.1    VISUALIZATION OF HETEROGENEITY

The Dirichlet non-iid model serves as our primary means to simulate realistic FL scenarios. Throughout this paper, we extensively explore the effects of varying the Dirichlet factor $\alpha$ and examine how our algorithms perform under different degrees of data heterogeneity. In Figure 10, we

present a visualization of the class distribution in the FedCIFAR10 dataset. We visualize the first 10 clients. This illustration clearly demonstrates that a smaller $\alpha$ results in greater data heterogeneity, with $\alpha = 1000$ approaching near-homogeneity. To further our investigation, we conduct thorough ablation studies using values of $\alpha$ in the range of $[0.1, 0.3, 0.5, 0.7, 0.9, 1.0]$. It is important to note that an $\alpha$ value of 1.0, while on the higher end of our test spectrum, still represents a heterogeneous data distribution.

### B.1.2 INFLUENCE OF HETEROGENEITY WITH NON-COMPRESSED MODELS

In our previous analyses, the impact of sparsified models with a sparsity factor $K = 10\%$ was illustrated in Figure 2, and the effects of quantized models were depicted in Figure 6. Extending this line of inquiry, we now present additional experimental results that explore the influence of data heterogeneity on models with $K = 50\%$ and those without compression, as shown in Figure 11. Our findings indicate that while model compression can result in slower convergence rates, it also potentially reduces the total communication cost, thereby enhancing overall efficiency. Notably, a Dirichlet factor of $\alpha = 0.1$ creates a highly heterogeneous setting, impacting both the speed of convergence and the final accuracy, with results being considerably inferior compared to other degrees of heterogeneity.

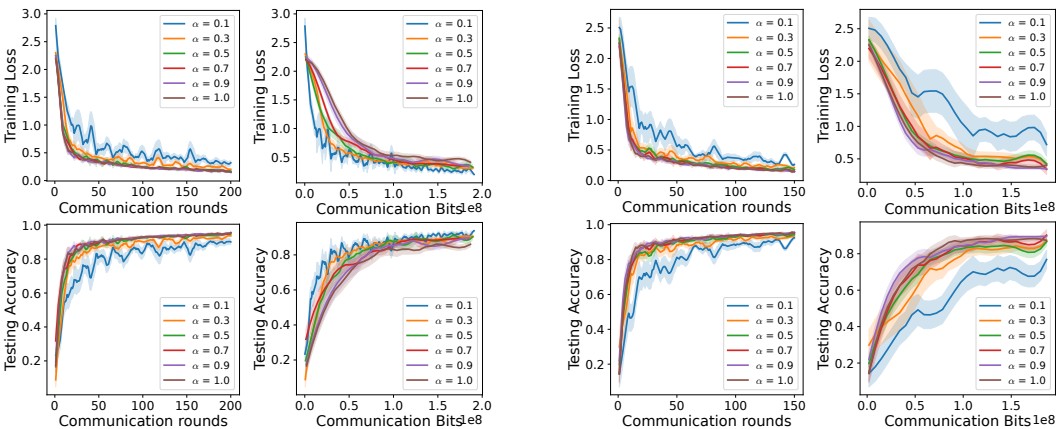

Figure 11: Exploration of variations in loss and accuracy across diverse sparsity ratios, communication rounds, and communicated bits is depicted through our figures. The first set of four figures on the left showcases results obtained with a sparsity ratio of $K = 50\%$. In contrast, the corresponding set on the right, consisting of another four figures, represents scenarios where $K = 100\%$, indicative of scenarios without model compression.

### B.2 COMPLEMENTARY QUANTIZATION RESULTS

### B.2.1 ADDITIONAL QUANTIZATION RESULTS ON FEDMNIST

In Figure 6, we presented the quantization results in terms of communicated bits. For completeness, we also display the results with respect to communication rounds in Figure 13.

### B.2.2 QUANTIZATION ON FEDCIFAR10

Previously, in Figure 5, we detailed the outcomes of applying quantization to the FedMNIST dataset. This section includes an additional series of experiments conducted on the FedCIFAR10 dataset. The results of these experiments are depicted in Figure 14. Consistent with our earlier findings, we observe that quantization considerably reduces communication costs with only a marginal decline in performance.

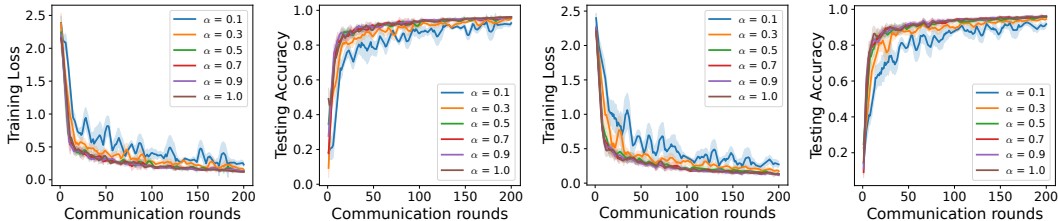

Figure 13: FedComLoc utilizing $Q_r(\cdot)$ with a fixed $r$ value of $8$ (as shown in the left figure) and $16$ (in the right figure) with respected to communication rounds. We conduct ablations across various $\alpha$.

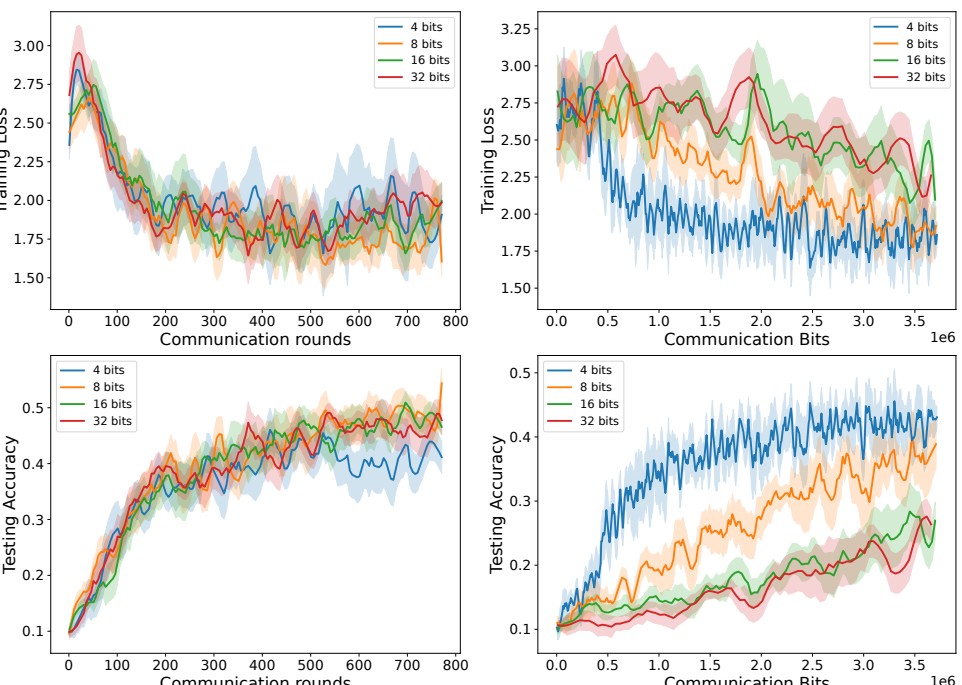

Figure 14: FedComLoc with $Q_r(\cdot)$ on CIFAR10.

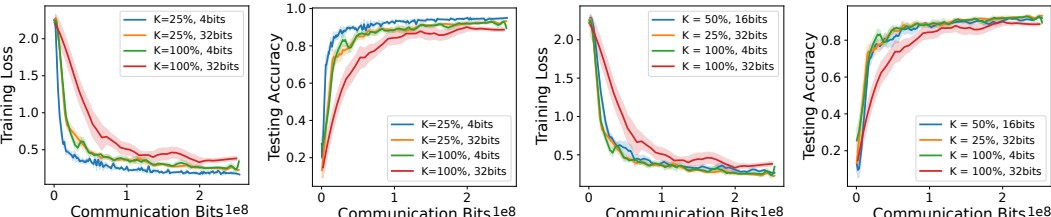

Figure 15: Comparison with double compression by sparsity and quantization.

## B.3 DOUBLE COMPRESSION BY SPARSITY AND QUANTIZATION

In Sections 4.1 and 4.4, we individually investigated the effectiveness of sparsified training and quantization. Building on these findings, this section delves into the combined application of both techniques, aiming to harness their synergistic potential for enhanced results. Specifically, we first conduct Top-K sparsity and then quantize the selected Top-K weights. The outcomes of this exploration are depicted in Figure 15. The left pair of figures illustrates that applying double compression with a higher degree of compression consistently surpasses the performance of lower compression degrees in terms of communication bits. However, the rightmost figure presents an intriguing observation: considering the communicated bits and convergence speed, there is no distinct advantage discernible between double compression and single compression when they are set to achieve the same level of compression.

