# OpenReview forum: "FedComLoc: Communication-Efficient Distributed Training of Sparse and Quantized Models"
_ICLR.cc/2025/Conference — ICLR 2025 Conference Withdrawn Submission_

### Official Review · Reviewer_6eQN · 2024-10-30

**Soundness:** 2
**Presentation:** 2
**Contribution:** 1
**Rating:** 3
**Confidence:** 4

**Summary:**

In this work the authors suggest to incorporate compression, via sparsification (TopK) and (random) quantization, into the established Scaffnew algorithm of Mishchenko et al. (2022); where the latter considerably advanced the reduction of communication complexity in federated learning. The integration of compressed communication into Scaffnew is studied in either of client-to-server, server-to-client, and client-storage possibilities; and experimentally verified using MNIST and CIFAR10 datasets in the non-iid local datasets scenario.

**Strengths:**

Clarity:
- The paper is overall well written and organized.
- The experiments' accompanied details and explanations are overall well presented.
- The ablation study is quite comprehensive.

**Weaknesses:**

Significance & Quality:
- Compression in FL has been extensively studied. It is  of lessen motivation to incorporate it into multiple SGD local iterations, what already by itself relaxes the communication overhead compared to single SGD local iteration, as communicating the local gradients to the server occurs less often.
- As stated by the authors in lines 177-179, this work's integration of compression in Scaffnew is merely heuristic and solely provides numerical evaluations for CompressedScaffnew (Condat et al., 2022); where the latter studies the theoretical aspects of general lossy compression integrated into Scaffnew in convex settings.
- The majority of FL papers with provable convergence guarantees employ convex settings in their theoretical evolutions and non-convex ones in their experimental studies; having deep neural networks fit into that regime.
- The CompressedScaffnew presents the idea of compressed Scaffnew, provides analytical study under the convex setting, and presents simulations that, unusually to their respective related works, covers only simplistic logistic regression model rather than diverse deep learning architectures. The significance of this paper is thus equivalent to the importance of a full-length comprehensive experiments section of the work CompressedScaffnew; i.e., one that includes simulations on general neural networks beyond the simplified logistic regression model. As a result, the merit of this paper to a conference such as ICLR is of minor contribution.

Originality:
- The idea learned in this paper is not new and was already presented, and also analytically analyzed, in CompressedScaffnew (Condat et al., 2022). The numerical simulations of this idea are claimed to be the novelty of the presented work, and by themselves are insufficient.
- The authors further claim in lines 169-171 that the studies of CompressedScaffnew (Condat et al., 2022) are not practical as they require
shared randomness. Yet, a bulk of works studying compressed FL utilize pseudo-random methods upon which sharing a common seed can overcome the necessity of shared randomness.

**Questions:**

1. In line 14 and in line 24 when you write 'heterogeneous clients' and 'heterogeneous settings', respectively,  it is unclear if you mean to heterogeneity in data or local hardware or both.
2. In lines 28-29: "Privacy concerns and limited computing resources on edge devices often make centralized training impractical, where all data is gathered in a data center for processing." Maybe you mean "Privacy concerns and limited computing resources on  a data center often make..."?
3. In lines 38-39: "Our primary objective is to solve the problem (ERM) and deploy the optimized global model to all clients.". Actually, in FL, it is often the server who wishes to obtain the optimal model rather than the local users.  The users are contributing their local private datasets to the process of learning, "serving" the centralizing server.
4. In line 40 there is a typo, it should be 'is' instead of 'are'.
5. In lines 54-55: "Quantization is another efficient model compression technique..., though its application in heterogeneous settings is limited". Quite a harsh statement. Do you have any references supporting it?
6. In lines 71-74 you claim that FedComLoc is specially designed for heterogenous environments. My question is why do you claim that, as your adopted compression methods, encompassing TopK and quantization, are generic tools in compression; and furthermore, your numerical evolutions involve the non-iid local datasets scenario, as in standard FL experimental studies.
7. In lines 75-78 you mentioned that the integration of compression communication into Scaffnew is studied in either of client-to-server, server-to-client, and client-storage possibilities. Actually, as also covered by the majority of compressed FL works, it is the client-to-server communication bottleneck that is the most crucial one to be relaxed.
8. Algorithm 1 is almost unreadable without being closely familiar with Scaffnew. For the paper to be a stand-alone one, you should specify in the accompanied text the not-intuitive-usages therein; e.g., control variates, the role probability, etc.
9. In theoretical studies of compressed FL, it is typically revealed that the integration of compression slows downs the convergence rate obtained without it. Can you explain how in the rightmost column of Fig. 3 the reversed is evidenced?
10. In lines 365-366: "Observe the accelerated convergence of sparsified models in terms of communicated bits...". It is not clear how this is being calculated. That is, to translate K into bits one can do, e.g., if K=10% set  R=0.1*b when b it the used-bits in full-precision, mostly 32 or 64. What the x-axis of 'Communication Bits' measures in your case?
11. It would be interesting to compare the performance of quantization and sparsification for the same bit rate...
12. In lines 370-371: "This indicates that sparsity training requires more data and benefits from either increased communication rounds...". Benefits? according to my understanding more communication rounds imply slower overall convergence (that takes longer time); which is not wanted. Can you explain that?

---

### Official Review · Reviewer_zB8T · 2024-11-02

**Soundness:** 2
**Presentation:** 2
**Contribution:** 2
**Rating:** 3
**Confidence:** 4

**Summary:**

This paper explores communication reduction techniques for a federated optimization method called Scaffnew. The proposed approach demonstrates that Scaffnew can be combined with various communication reduction techniques on both the local and global sides. Empirical results illustrate the effectiveness of FedComLoc in reducing communication overhead while maintaining comparable performance.

**Strengths:**

1. The paper is clearly well-written, concise, and easy to follow.

**Weaknesses:**

1. My major concern is the novelty and technical contribution of this paper. Model compression techniques, such as top-k and quantization, are already widely used and well-established. Integrating these compression methods with an FL algorithm appears to be an incremental contribution. While this approach does address the communication cost challenges in Scaffnew, it is not immediately clear to me how applying model compression introduces new challenges or is non-trivial. Therefore, the technical contribution seems relatively weak to me.

2. It appears that the proposed algorithm and experiments are conducted under a partial participation setting in FL. This could lead to potential “asynchronous” issues in FedComLoc-Global: since there is no model initialization step in FedComLoc, a client that has not participated in the previous $t-1$ steps would begin local training in the $t$-th step with an outdated model. This lack of updated model initialization may result in poorer convergence, particularly when only a small fraction of clients participate in the training.

3. There are several issues with the presentation of the experimental results:

a. The caption of the subfigures in Figure 1 mentions sparsity, but the curves also represent sparsity.

b. The results in the table in Figure 6 conflict with those in the subfigure within the same figure.

c. The caption of Figure 8 states that there is a comparison with FedDyn, but the subfigures do not include this baseline.

d. What is the purpose of K=100% (no sparsity) in Figure 8? It seems this is intended to compare Scaffnew with FedAvg and Scaffold.

**Questions:**

Please refer to the weaknesses section for details.

---

### Official Review · Reviewer_7m4M · 2024-11-03

**Soundness:** 1
**Presentation:** 2
**Contribution:** 1
**Rating:** 3
**Confidence:** 4

**Summary:**

This paper presents an empirical study on a new algorithm, FedComLoc, which extends Scaffnew by integrating compression techniques: TopK and quantization. Three settings of the proposed algorithm are evaluated: (i) compressing the communication from client to the server; (ii) compressing the local model itself; and (iii) compressing the communication from the server to each client. The paper reports the empirical performances for these configurations by using FedMNIST and FedCIFAR10 with varying degrees of heterogeneity.

**Strengths:**

- The paper reports and discusses the empirical performances of FedComLoc in different configurations and hyperparameter settings, including heterogeneity, quantization bits, and TopK sparsities.
- The paper presents some interesting observations based on the numerical results.

**Weaknesses:**

- The scope of the paper is narrow, focusing solely on one algorithm, Scaffnew, with a basic integration of existing compression schemes. The paper lacks justification for why Scaffnew was chosen over other potential algorithms.
- The paper provides an insufficient review of existing communication-efficient FL approaches (e.g., FedEF [1]). These existing approaches were also missing in the numerical experiment.
- The experimental setup is not extensive, relying only on image datasets and simple models: MLPs and CNNs. Given the paper's empirical focus without theoretical analysis or in-depth discussion, it is challenging to generalize the findings. Moreover, the numerical comparison is limited and lacks breadth.
- The paper offers no new insights or findings beyond empirical results and observations. The main conclusion appears to be: "we can apply compression schemes to Scaffnew."

**Reference**
1. Li and Li. Analysis of error feedback in federated non-convex optimization with biased compression: Fast convergence and partial participation. ICML, 2023.

**Questions:**

1. Why was Scaffnew chosen specifically for this study? What unique characteristics of Scaffnew make it suitable for the proposed compression techniques compared to other algorithms?
1. Do the authors expect the empirical observations to generalize to other algorithms using the same compression schemes? If so, could the paper include a discussion on the expected performance of these compression schemes when applied to other popular algorithms?
1. If the goal is to develop a communication-efficient algorithm that outperforms the state-of-the-art (SOTA), what are the current SOTA methods in communication efficiency? The proposed method integrates compression schemes with one particular algorithm. Are there other approaches that might achieve comparable communication efficiency in federated learning? A review and comparison with SOTA methods could strengthen the context of this work.
1. Are the sub-captions in Figure 1 accurate, or could there be some typos that need correction? Please confirm and revise if necessary to ensure clarity.

---

### Note · Authors · 2024-11-18

I have read and agree with the venue's withdrawal policy on behalf of myself and my co-authors.